# Generalizing the isothermal efficiency by using Gaussian distributions

**Thomas D. Schneider** *

National Institutes of Health, National Cancer Institute, Center for Cancer Research, RNA Biology Laboratory, Frederick, MD, United States of America

* schneidt@mail.nih.gov

## Abstract

Unlike the Carnot heat engine efficiency published in 1824, an isothermal efficiency derived from thermodynamics and information theory can be applied to biological systems. The original approach by Pierce and Cutler in 1959 to derive the isothermal efficiency equation came from Shannon's channel capacity of 1949 and from Felker's 1952 determination of the minimum energy dissipation needed to gain a bit. In 1991 and 2010 Schneider showed how the isothermal efficiency equation can be applied to molecular machines and that this can be used to explain why several molecular machines are 70% efficient. Surprisingly, some macroscopic biological systems, such as whole ecosystems, are also 70% efficient but it is hard to see how this could be explained by a thermodynamic and molecular theory. The thesis of this paper is that the isothermal efficiency can be derived without using thermodynamics by starting from a set of independent Gaussian distributions. This novel derivation generalizes the isothermal efficiency equation for use at all levels of biology, from molecules to ecosystems.

**Data Availability Statement:** All relevant data are within the paper and its Supporting information files.

**Funding:** This research by TDS was supported by the Intramural Research Program of the NIH (https://www.nih.gov/), National Cancer Institute

## 1 Introduction

"...anything found to be true of *E. coli* must also be true of Elephants." – Jacques Monod and François Jacob [1, 2].

"...it is clear that the prime intellectual task of the future lies in constructing an appropriate theoretical framework for biology." – Sydney Brenner [3]

Carnot derived a nonisothermal equation for the maximum efficiency of a heat engine, a device that uses a heat source at temperature $T_{\text{hot}}$ and a heat sink at $T_{\text{cold}}$:

$$\eta_{\text{Carnot}} = \frac{T_{\text{hot}} - T_{\text{cold}}}{T_{\text{hot}}} \tag{1}$$

[4, 5]. Because biological systems usually function at a single temperature, $T_{hot} = T_{cold}$ so $\eta_{\text{Carnot}} = 0$, making the Carnot efficiency inappropriate to use to investigate most of biology. This was recognized by Jaynes [6] who attempted to describe the efficiency of muscle anyway by using 'effective temperatures'. As has been confirmed by modern experiments [7, 8], Jaynes knew that the maximum muscle efficiency is about 70% [9], and since muscles work at

(https://www.cancer.gov/), Center for Cancer Research (https://ccr.cancer.gov/). The funders had no role in study design, data collection and analysis, decision to publish, or preparation of the manuscript.

**Competing interests:** The authors have declared that no competing interests exist.

approximately $T_{cold} = 300K$, he inferred that $T_{hot} = 1000K$. Fahrenheit 451, the famous book by Ray Bradbury [10], suggests that paper spontaneously ignites at 506K [11, 12], well below this $T_{hot}$ number. The result implies that muscles would be hot enough to burst into flames when they function, which is absurd since the muscle proteins actin and myosin denature well below boiling water at 100C = 373K [13, 14]. Because of this problem, Jaynes pointed out that muscle is not a heat engine, yet he still tried to use the Carnot equation. Likewise, modern work on 'efficiency at maximum power' is also still based on Carnot and so is not relevant to most biological systems [15, 16]. Fortunately, instead of attempting to shoehorn the biology to fit the Carnot efficiency, one can derive an efficiency that applies at a single temperature (see Eq (9)). Like the nonisothermal Carnot efficiency, this isothermal efficiency is also derived from the second law of thermodynamics [17–19].

After Shannon published his famous paper on information theory in 1948 [20], he followed in 1949 with a short but brilliant paper on the geometrical representation of coding systems [21]. In this paper Shannon derived the maximum information that can be sent over a communications channel, the channel capacity. Ten years later in 1959 Pierce and Cutler built on these concepts to define an efficiency for satellite communications [22]. Another 30 years later, in 1989, I found that the channel capacity equation can be generalized to apply to molecular machines, molecules that dissipate energy to select between two or more states, as defined in reference [17]. Surprisingly, the channel capacity is closely related to the second law of thermodynamics for processes that function at one temperature [18]. These ideas allowed me to derive the isothermal efficiency for molecular machines and to explain why the DNA binding protein EcoRI and the molecular light switch rhodopsin also operate near 70% efficiency [19].

In this paper I show how to generalize the channel capacity and isothermal efficiency equations even further so that they can be applied to a much wider range of biological problems such as entire ecosystems. The basic idea is that one can construct these equations starting from a set of independent Gaussian distributions instead of from electrical signals [21] or thermodynamics [17–19]. The paper begins in Section 2 with a review of the relevant concepts developed previously and then gives the new derivation in Section 3.

The purpose of this paper is to generalize the efficiency mathematics for application across biology. The key idea that many biological systems approach $\ln 2 \approx 70\%$ efficiency, and its explanation, is already published [19] and has been used to explain why restriction enzymes frequently have binding sites that are six bases long: they likely use the 24 dimensional Leech lattice for coding [23]. Extensive data will be presented elsewhere, so the 70% observation does not warrant further support here. Also, I intend this paper to be readable by biologists and others interested in the topic so it briefly explains basic information theory but does not give detailed mathematical proofs. The famous mathematics expositor Paul R. Halmos said

> The best notation is no notation; whenever it is possible to avoid the use of a complicated alphabetic apparatus, avoid it. . . .fall back on symbolism only when it is really necessary.

[24] (section 15). Those interested are invited to write a formal mathematical proof.

## 2 Channel capacity and efficiency in biological systems

### 2.1 What is channel capacity?

For Shannon the derivations began with the assumption that a message will contain a number of independent symbols, each symbol represented by a specific voltage. Problematically, pristine voltage pulses become corrupted by thermal noise in the communications channel such as a wire; this is the hiss that one may hear on AM radio. Classical thermal noise, explained in

1928 by Nyquist [25] and Johnson [26], results from summing many small independent random impacts, so according to the central limit theorem of statistics it has a Gaussian distribution. In information theory this is known as Additive White Gaussian Noise (AWGN). Shannon recognized that this is the worst possible noise and he focused his efforts on overcoming it. He did so by an elegant trick. When two independent Gaussian distributions are graphed orthogonally, they form a circular (Rayleigh) distribution [17, 27]. Three independent Gaussian distributions combine to form a sphere, and the probability density in spherical shells is the Maxwell gas distribution. As one adds more Gaussian distributions, the result is still spherical in higher dimensions, and results in a $\chi^2$ probability distribution. As the dimension (number of independent Gaussians) increases, the $\chi^2$ distribution converges sharply to a thin spherical shell [5, 17, 28].

Shannon showed that any message can be represented as a series of voltage pulses and since these are independently chosen by the sender, the entire message can be represented as a point in a high dimensional 'coding' space. When this message is transmitted through a communications channel, thermal noise is added to each pulse. So a set of zero voltage pulses will be smeared out into a Gaussian distribution, as will the pulses of other initially precise voltages of the message. Because each pulse and its noise is independent of the other pulses, the receiver gets a point on the surface of a high dimensional sphere surrounding the original message point.

By prior agreement with the transmitter, the receiver can know all the allowed locations of transmitted messages in the high dimensional space, so when it gets a noise-distorted message it can determine the closest possible transmitted message. That is, given a point on one of the received message spheres, the center of the sphere can be determined in a process called 'decoding'. By substituting the closest possible transmitted message point for the received noisy point, the noise can be removed. The pure message is then given to the destination. Modern communications systems, such as cell phones, use these concepts to provide clear signals even though thermal noise is prevalent. This scheme only works if the spheres do not significantly overlap since overlaps would make decoding ambiguous. The probability of errors per message symbol can be driven as low as one may desire by increasing the number of dimensions to sharpen the spheres. So Shannon discovered that it is not necessary to increase the power to reduce the error rate [21].

A similar argument has been made for molecular machines [17]. In an electrical circuit, voltage squared is proportional to the energy dissipated from a resistor per second. Likewise, velocity squared is proportional to the energy of a moving body. So the mechanical equivalent of Shannon's voltage pulses is the velocity of sets of atoms that work together in a molecule. Considering the analogy of a lock is useful for visualizing this idea. A lock consists of a set of two-part pins that can move up and down (Fig 1). The pins have different lengths. When a key is inserted into the lock the pins are moved up and down and if it is the right key, the breaks between the parts align at the 'shear line', allowing the lock to open [29]. The pins in a lock can move independently, so they can represent different dimensions. I proposed that specific parts of molecules act like pins in a lock [17]. Gaussianly distributed thermal noise interacts with each pin of the molecule. So the noise impacting on a molecule can be represented as a sphere in a high dimensional space. The more 'pins' there are in the molecule, the more distinct that spherical state can be from other spherical states [5, 17, 21, 28].

So now we have two models, Shannon's voltage space [21] and my molecular velocity space [17]. In both cases, the total energy available (noise plus power) determines a large sphere around the smaller message or molecular state spheres. Shannon realized that the smaller thermal noise spheres could pack together inside the larger sphere (Fig 2) and that by dividing the volume of the larger sphere by the smaller sphere volume one could determine the number of

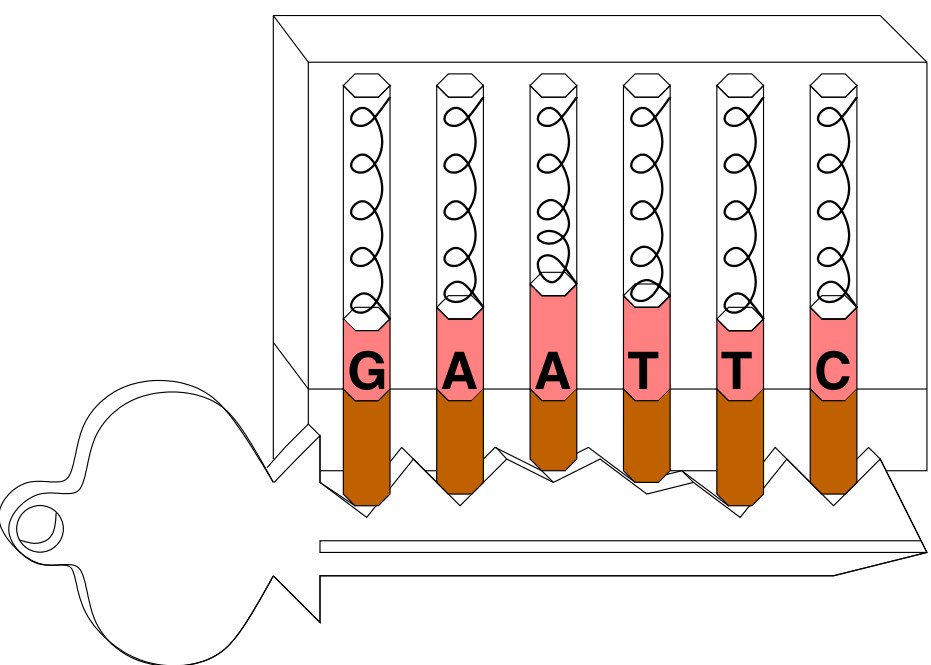

**Fig 1. Lock-key model for the EcoRI restriction enzyme.**

possible messages. By taking the logarithm of this number he derived the channel capacity:

$$C = W \log_2 \frac{P + N}{N} \quad \text{(bits per second)} \tag{2}$$

where $P$ is the signal power absorbed from the message and then dissipated by the receiver, $N$ is the noise interfering with the receiver, and $W$ is the bandwidth (the range of frequencies used in the communication). An equivalent formula was developed for molecules, and in this case $W$ is replaced by the number of independent 'pins', $d_{\text{space}}$ [17].

Shannon then showed that a communication system cannot send data at a rate higher than the channel capacity, but if the rate is less than or (surprisingly) equal to the capacity, the error may be made as small as desired [21]. The equivalent theorem for molecules making state selections is that as long as the molecular state capacity is not exceeded, molecules may make as few errors as necessary for survival of the organism they support. ('Desire' is not an appropriate way to think about naturally occurring biological systems since they evolve and are not designed [30] the way communications systems are).

## 2.2 Measuring molecular efficiency

Many genetic controls are accomplished by proteins that bind to specific patterns in DNA. The information in these patterns can be computed [31, 32] and displayed graphically using sequence logos [33] (Fig 3).

How is this information related to the binding energy $\Delta G$? As we will see below, the Second law of Thermodynamics provides a way to determine the number of bits that could be gained for the binding energy dissipation [18]. So we can divide the information in a DNA protein binding pattern by the information that could have been attained from the energy dissipation. This is the molecular efficiency. Since the molecules work at one temperature, it is also an isothermal efficiency.

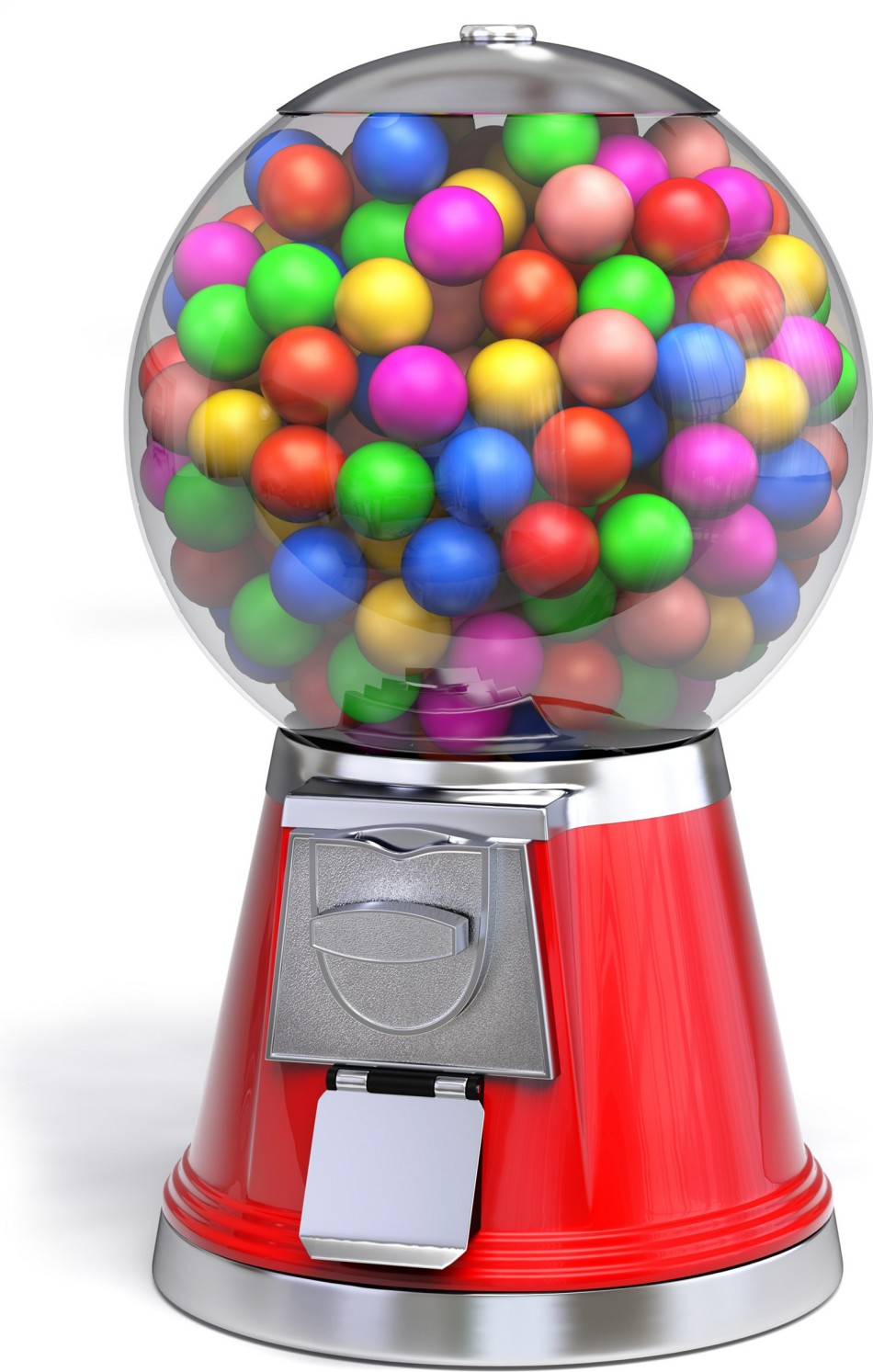

**Fig 2. A gumball machine represents sphere packing in a high dimensional space.** "Gumball machine" http://www.
turbosquid.com/3d-models/3d-model-gum-gumball-machine/648046 by Guido Vrola Design 2016 http://vroladesign.
it/ is licensed under CC BY 4.0, reproduced with permission.

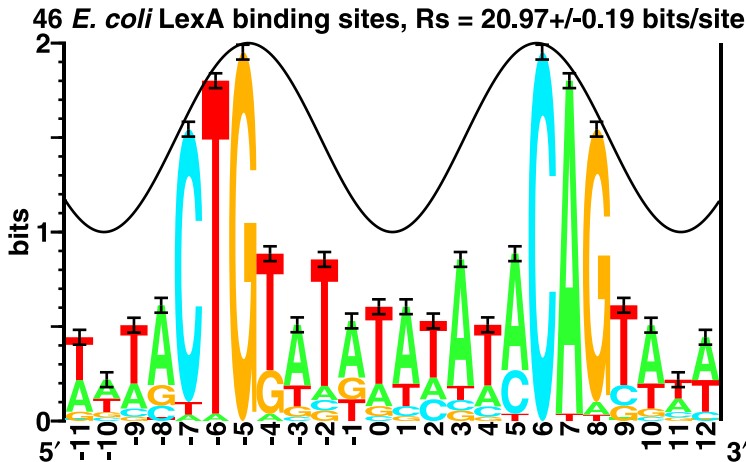

**Fig 3. Sequence logo for LexA binding sites [33, 34].** 23 experimentally demonstrated DNA binding sites for the LexA protein and their complementary sequences from the bacterium *E. coli* (Genbank accession NC_000913.3) were collected and aligned using Delila instructions [35] that precisely define the site locations (see https://alum.mit.edu/www/toms/lexa-inst.txt, and S1 Text lexa-inst.txt). The sequence logo shows stacks of letters corresponding to positions in the alignment. Within a stack, the height of a letter is proportional to the frequency of the base at that position. The entire stack height is the sequence conservation at that position, measured in bits [31]. I-beam error bars show the likely variation caused by the small sample size [31]. The sum of the heights of the logo stacks, the 'area' under the logo, is $R_{sequence} = 20.97 \pm 0.19$ bits per site which can be predicted from the size of the genome ($G$) and the number of sites ($\gamma$) using $R_{frequency} = -\log_2(\gamma/G)$ bits per site [31, 32]. The peak of the sine wave indicates where the protein faces the major groove of the DNA; intriguingly, this follows the sequence conservation shown by the logo [34, 36].

Along with this empirical computation, one can develop a theoretical equation for the efficiency (see Eqs (9) and (33)). This was first derived by Pierce and Cutler to determine the efficiency of satellite communications [22]. Apparently the isothermal efficiency equation was not used after Pierce and Cutler until it was applied by me to DNA binding proteins [19]. A clear case is EcoRI, a defense enzyme of the bacterium *Escherichia coli*. After a virus injects its DNA into a bacterium, the bacterial EcoRI molecules move along the DNA by Brownian motion and then bind to the 6 base-pair long viral DNA sequence 5′ G↓AATTC 3′ and cut both strands of the double helical DNA between the G and the A. This breaks the viral DNA into pieces and the virus is defeated [37]. The host DNA is not cleaved because it is methylated by another enzyme at the second A of the GAATTC sequence.

Before we can begin the calculation of EcoRI efficiency, it is necessary to take a small detour into basic information theory [38, 39]. A coin can sit stably on a table in only two possible states of heads or tails. This means that a coin can store $\log_2 2 = 1$ bit of information. Further extending this argument to choose one out of the four DNA bases (represented by the letters, A, C, G and T) one can first arrange them on the corners of a square. One can think of a bit as a knife slice that splits the set of possibilities in half. Two knife slices will distinguish the four bases as either being on top/bottom or left/right. This requires two independent choices or $\log_2 4 = 2$ bits of information per base. Base independence is an assumption, but it is reasonably well supported for DNA binding proteins [40]. For this example we also assume equiprobable DNA bases [31].

There are 6 successive bases in the EcoRI binding site, so the total information—which Shannon required to be additive for independent components [20]—is 2(*bitsperbase*) × 6(*basespersite*) = 12 (bits per site) [31].

EcoRI selects one state out of $4^6 = 2^{12} = 4096$ possible DNA sequences that are 6 bases long. In order to stick and remain in place, EcoRI dissipates energy when it binds to the DNA. The

question of the relationship between energy and information, exemplified by the action of EcoRI, has been a fundamental question in biology ever since Maxwell invented his demon [41].

In the case of a coin one may start at some height above a table, so it has some potential energy relative to the table surface. There will also be some kinetic energy in the coin if it is moving. Our goal is to set the coin on the table in a specific state—this is not a random flip. When we place the coin either heads or tails on the table, both the potential and the kinetic energy must be dissipated to the surroundings. If these were not dissipated, the coin would either not be resting on the table (potential) or it would bounce (kinetic).

If one starts a motionless coin 10 cm above the table, some potential energy must be dissipated to store one bit. If one were to start 20 cm above the table, then twice as much energy would have to be dissipated to still gain only one bit. Therefore the relationship between information and energy is an inequality. So, what is the minimum energy needed to store a bit?

To answer this important question we can use the Clausius inequality version of the Second Law of Thermodynamics:

$$dS \geq q/T \tag{3}$$

where $dS$ is the change of entropy of a system and $q$ is the heat put into the system at absolute temperature $T$ [42]. By setting $T$ to a constant, to represent isothermal biological conditions, I derived the minimum energy that must be dissipated from a system to gain one bit in the system:

$$E_{min} = k_B T \ln 2 \qquad \text{(joules per bit)} \tag{4}$$

where $k_B$ is Boltzmann's constant and $\ln 2$ sets the units to 'per bit' [18]. Historically, this same equation was first derived from the channel capacity by Felker in 1952 [18, 22, 43–46], not Landauer in 1961 [47] as is often assumed.

When EcoRI binds to DNA, it dissipates energy in going from anywhere on the DNA to its binding sites. The equilibrium constant has been measured as $K_{spec} = (1.59\pm0.14) \times 10^5$ [48]. The free energy, which gives the maximum amount of work that can be done by a chemical reaction, is

$$\Delta G^{\circ}_{spec} = -k_B T \ln K_{spec} \qquad \text{(joules per binding)}. \tag{5}$$

Combining this equation with Eq (4) gives the informational equivalent of the energy

$$R_{energy} \quad \equiv \quad -\Delta G^{\circ}_{spec}/E_{min} \tag{6}$$

$$= \quad \log_2 K_{spec} \qquad \text{(bits per binding)} \tag{7}$$

and we learn by inserting $K_{spec}$ that for this energy dissipation, EcoRI could have made 17.3 ±0.1 bits of decision.

But, as we saw earlier, the pattern in the DNA is only $R_{sequence} = 12$ bits per binding, so EcoRI has an efficiency of only

$$\begin{aligned} \epsilon_r &= \frac{R_{sequence}}{R_{energy}} \\ &= 12/(17.3 \pm 0.1) \\ &= 69.4 \pm 0.4\%. \end{aligned} \tag{8}$$

A similar calculation made for LexA using the area under the sequence logo (Fig 3) and measured binding constants [49] gives an isothermal efficiency of 0.73±0.02.

After muscle (0.68±0.09 [7, 8]), EcoRI (0.694±0.004) and LexA (0.73±0.02), another molecular example suggests that having $\sim 70\%$ efficiency represents a general rule. The protein molecule rhodopsin is found in the retina of the eye. Buried in the center of rhodopsin is the light sensitive pigment retinal, a form of vitamin A related to the $\beta$-carotene that gives some carrots an orange color. When a photon hits the retina and is absorbed by retinal, rhodopsin starts wiggling and then it can switch to a new state to record the arrival and absorption of the light [50, 51]. However, it switches only 66±3% of the time [19].

A 70% efficiency can be explained using the isothermal efficiency equation derived from the channel capacity, Eq (2):

$$\epsilon_t = \frac{\ln\left(\frac{P}{N} + 1\right)}{\frac{P}{N}} \tag{9}$$

where $P$ is the power dissipated (energy per binding for example), $N$ is the thermal noise that disrupts the molecule during binding and the subscript $t$ represents the theoretical bound as opposed to a measured 'real' efficiency $\epsilon_r$ (Fig 4) [19, 52]. The channel capacity theorem says that no system can exceed this boundary,

$$\epsilon_r \leq \epsilon_t. \tag{10}$$

The efficiency of a biological system will tend to a maximum. For EcoRI, excess energetic contacts that don't specify information will be mutated away so according to Eq (8) the efficiency will rise. Likewise, it is imperative for animals that have vision to capture the maximum number of photons to survive. Yet no more than 70% of the photons actually absorbed by rhodopsin are detected. The reason for an upper bound of 70% is not obvious.

However, when the $P/N$ ratio is 1, the efficiency according to Eq (9) is $\ln 2 \approx 0.69$. Because Eq (9) is a monotonically decreasing upper bound, the observation of a 70% efficiency maximum could be explained instead as a lower bound on $P/N$. It turns out that when $P = N$, the

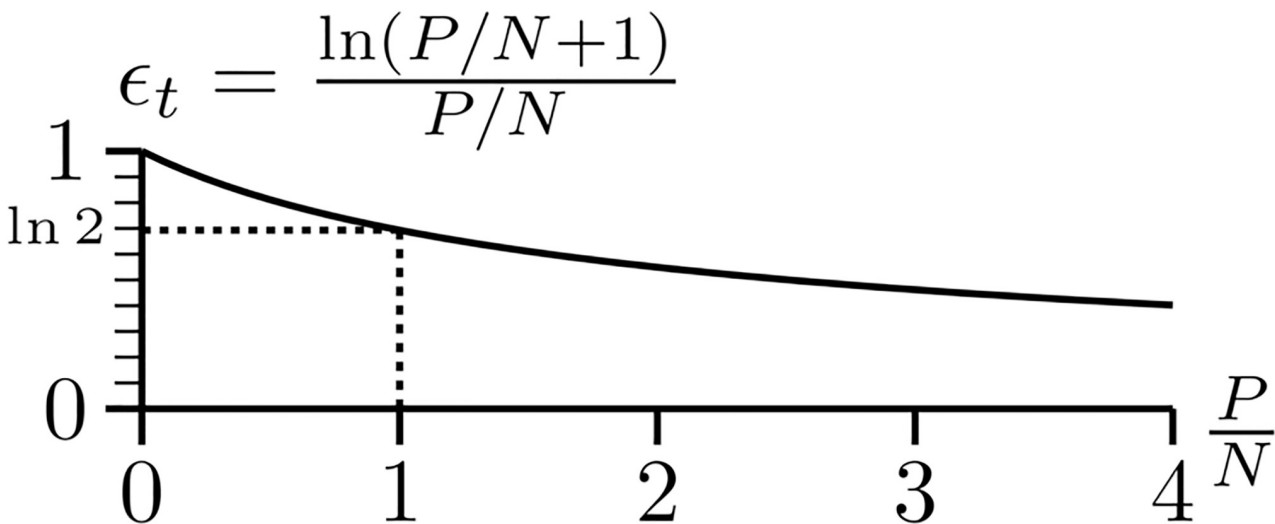

**Fig 4. The isothermal efficiency curve (Eq (9)) derived either from information theory and thermodynamics [19, 22, 52] or Gaussian distributions (this paper).** Shannon's channel capacity theorem [21] ensures that no system can lie above the curve.

spherical states of the molecules are just touching [19, 52], as shown in Fig 2. The implication is that the states of the molecule are distinct, as is well known for EcoRI and rhodopsin.

By reversing the order of this argument we can see the biological significance of the 70% results. We start from the fundamental requirement that EcoRI sites be distinct from other sites in the genome and for rhodopsin's dark versus light states to be distinct. The methylase enzyme that is paired with EcoRI only protects the DNA sequence GAATTC by methylation, so by evolving to satisfy the separation of states requirement EcoRI will not chew up the rest of the unprotected genome. For rhodopsin the state separation requirement prevents one from continuously seeing flashes of light, which would make one blind in the extreme. The state separation requirement implies that the state spheres not intersect and so $P/N > 1$ and therefore $\epsilon_t < \ln 2 \approx 0.69$ by Eq (9). The observed 70% efficiency comes from the requirement that biological states be distinct in the presence of unavoidable high dimensional Gaussian noise that makes those states be spherical.

## 3 Developing a new derivation of the isothermal efficiency

### 3.1 70% efficiency also appears in ecology

Ecologists are concerned with the distribution of species in an ecosystem. If each species had the same number of individuals then the distribution would be flat and this is called 100% 'even'. Evenness could be calculated many ways and ecologists disagree on which of at least 14 proposed measures methods is best [53]. However, Evelyn C. Pielou proposed one such measure based on information theory [54]. Her measure uses Shannon's uncertainty [20]:

$$H = -\sum_{i=1}^{M} P_i \log_2 P_i \tag{11}$$

where $P_i$ is the probability of the $i^{th}$ species out of $M$ species in an ecosystem. The highest possible value of $H$,

$$H_{\max} = \log_2 M \tag{12}$$

occurs when all the probabilities are equal. This would be the most 'even' distribution of species, so Pielou proposed to measure ecological 'evenness' as:

$$J = H/H_{\max} \tag{13}$$

which gives evennesses between 0 and 1. The advantage of this evenness measure over all the others is that it is related to Shannon's channel capacity theorem and the isothermal efficiency. In her 1966 paper, Fig 3 shows $J$ peaking near 70% for ground vegetation. Can the thermodynamic explanation that applies to molecular machines be generalized so that it also applies to an ecosystem?

### 3.2 High dimensionality and noise lead to hyperspheres

The goal of this paper is to generalize the theory to accommodate molecular to macroscopic observations of 70% efficiency or evenness. We begin by noting that biological systems in general are described by many independent numbers. EcoRI, rhodopsin, actin and myosin are large molecules with potentially many 'pins', and a particular species of plant lives in an ecosystem niche with many conditions required for survival such as amount of sunlight, water and soil pH. That is, at all levels biological systems are high dimensional.

There is noise impacting each of the dimensions of a biological system. The noise comes in collisional impulses at the molecular level and fluctuating wind, rain and so on at the

ecosystem level. The combination of many small impulses tends to a Gaussian distribution according to the central limit theorem. An example is the approximately Gaussian distribution of heights at which different warbler species nest in conifer trees [55].

So we have the two factors needed to generate spheres in a high dimensional coding space: high dimensionality and independent Gaussian distributions. That is, biological states can, in general, be represented as hyperspheres. They become more sharply defined as the dimension increases, with the density concentrating on the surface [17], and this allows for the states to be more distinct. Biological systems can attain distinct states by increasing the dimensionality, so they will likely evolve high dimensional states. In the world of communications, if some variables are correlated, one can choose those that are independent [21]. In a biological system it is possible for the organism to evolve many independent components (like pins in a lock), and it is to the advantage of the organism to do so because that leads to sharply defined high dimensional states.

We can represent the noise along one dimension $y_i$ by a Gaussian probability distribution:

$$p(y_i) = \frac{1}{\sigma\sqrt{2\pi}} e^{-(y_i - \mu_i)^2/2\sigma^2} \tag{14}$$

where $\sigma$ is the standard deviation and $\mu_i$ is the mean (center) of the distribution [21]. In $D$ dimensions, $i = 1, \ldots, D$, we can integrate in spherical shells and the density is found to be

$$f_D(r) = \frac{r^{D-1} e^{-r^2/2\sigma^2}}{\Gamma\left(\frac{D}{2}\right) \sigma^D 2^{\frac{D}{2}-1}} \tag{15}$$

which is a $\chi^2$ distribution in $x = r^2/\sigma^2$ with $D$ degrees of freedom (See Appendix 3 of [17]). As the dimensionality increases, this distribution becomes more sharply peaked so we can model the states as hyperspheres with a radius of $\sigma\sqrt{D-1}$ [17].

People often speak of independent and identically distributed random variables (IID). This is too restrictive for the general biological problem of biological states. Each independent Gaussian distribution can be normalized to have the same standard deviation, but with different means, so they are not identical. This creates normalized Gaussian spheres in the coding space. For example, the mean and standard deviation for two different distributions that affect the ecological niche of a bacterium in the digestive system can have unique units such as Mg concentration and temperature. By dividing both the mean and standard deviation of each distribution, $i$, by its corresponding standard deviation, one converts a normal $N(\mu_i, \sigma_i)$ distribution (with mean $\mu_i$ and standard deviation $\sigma_i$), to a $N(\mu_i/\sigma_i, 1)$ distribution. In high dimensions the collection of all such independent distributions are then spheres with radius 1 at different locations in the space. In more formal terms, a biostate can be represented as a vector of real valued numbers $X_1, \ldots, X_D$ and the 'noise' as a normalized Gaussian vector $Z_1, \ldots, Z_D$, such that the observed state is $Y_i = X_i + Z_i$ (for $i = 1, \ldots, D$), which is a sphere of radius 1. As the dimensionality $D$ increases, the probability density concentrates on the surface in a thin shell by Eq (15).

The dimensionality and packing arrangement of these spheres—the coding (Fig 2)—is an open problem in biology. For example, we have recently shown how to determine the dimensionality of DNA binding proteins. Surprisingly EcoRI and other 6-base cutting restriction enzymes function in 24 dimensional space and probably use the famous Leech lattice for their sphere packing [23].

### 3.3 Define noise $N$ in terms of the standard deviation of a Gaussian, $\sigma$

In Shannon's communications model, the total noise $N$ is determined from the sum of the noise in each dimension. In a thermodynamic system, the dimensions are the independent degrees of freedom and according to the equipartition theorem of thermodynamics each carries an average energy of $\frac{1}{2}k_{\mathrm{B}}T$, so the total noise energy is:

$$N = \frac{1}{2}k_{\mathrm{B}}TD \qquad (16)$$

where $D$ is the dimensionality, $T$ is the absolute temperature and $k_{\mathrm{B}}$ is Boltzmann's constant [6, 17]. In an electrical system, the voltage squared is proportional to the power (energy per time) and in a mechanical biological system, the velocity squared is proportional to the energy per molecular machine operation [17]. Starting from the combined Gaussian density Eq (15), The radius of a Gaussian noise hypersphere is:

$$r = \sigma\sqrt{D-1} = \sqrt{N} \qquad (17)$$

(see equation 25 of [17]).

Again, note that if different dimensions have different units, then each dimension can be normalized by dividing by the standard deviation. That is, scaling along the different dimensions is irrelevant because one can normalize to make the spheres have the same radii. Also, if variables are correlated then, following Shannon, select the dimensions that are independent [21], as in principal component analysis.

So if we start from the premise that a biological system has many dimensions (independently varying components) and each dimension is disturbed by Gaussian noise, then the state of the biological system can be represented as a sharply defined sphere in a high dimensional space $D$ with a radius $\sqrt{N}$.

### 3.4 Define power $P$ as the energy dissipation from a system in the relevant biological time interval

A biological system can have many states, each represented by a hypersphere. The total number of accessible states is limited by the energy dissipation or power $P$. For EcoRI this is the so-called 'binding energy' and for rhodopsin this is the energy fleetingly remaining in the protein structure after that structure has been partially denatured by the photon. For molecular machines the time interval is the molecular machine operation, as previously discussed [17].

When this power $P$ is dissipated, it allows the system to move in the coding space from one hypersphere state to another. If the hyperspheres do not intersect, then the distance moved in the space $\sqrt{P}$ must at least exceed the radius of a hypersphere, so $\sqrt{P} > \sqrt{N}$ and so $P/N > 1$. (See [19] for more detailed proofs.) If we could apply the thermodynamically-derived Eq (9), then the efficiency would be near $\ln 2 \approx 70\%$. So how do we derive the efficiency equation without depending on thermodynamics?

### 3.5 Generalized isothermal efficiency

A vector for the power $P$ can point in any direction in the coding space. So the power inscribes a sphere of radius $P$ in the hyperspace into which must fit the centers of all of the noise hypersphere states that are accessible given that power. Some noise hyperspheres will peek outside the power sphere, extending the volume covered. To see how much, we must consider an odd property of high dimensional spaces that Shannon used in his proof of the channel capacity theorem [21]. Consider a power dissipation in a $D = 100$ dimensional space. The power allows

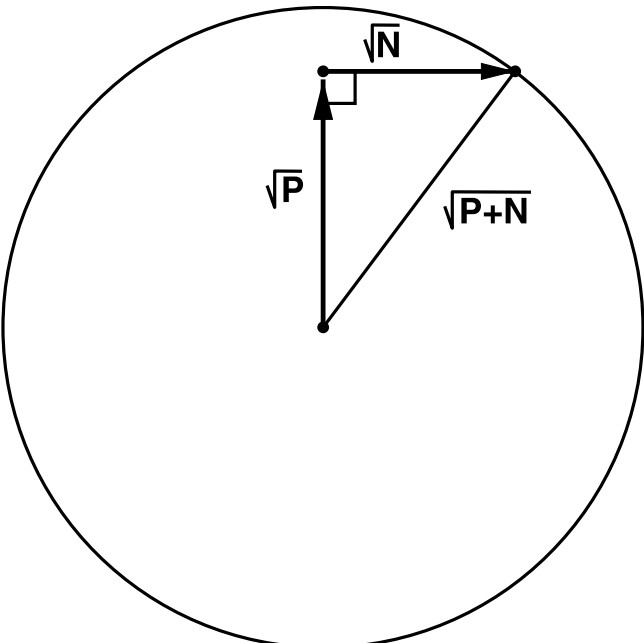

**Fig 5. Geometry of hyperspace vectors.**

the system to move in a particular direction to select a particular noise-sphere state. By the definition of a high dimensional space, 1% of the noise is in the power dissipation direction while 99% of the noise is at right angles, so the noise in the power dissipation direction is only 1/100[th] of the total noise. This is negligible from the viewpoint of a practical engineer like Shannon. So we effectively have a power vector with magnitude $\sqrt{P}$ at right angles to the noise vector with magnitude $\sqrt{N}$. The hypotenuse of the resulting triangle is, thanks to Pythagoras:

$$r_{\text{before}} \quad = \quad \sqrt{\sqrt{P}^2 + \sqrt{N}^2} \tag{18}$$

$$= \quad \sqrt{P + N} \tag{19}$$

(Fig 5) and this is the radius of a sphere that encloses all of the noise sphere states. Since this large sphere represents the state just before energy dissipation, I call it the *before* sphere. It is represented by the transparent shell surrounding the gumballs in Fig 2. Likewise, after dissipation the system has settled into one of the smaller Gaussian noise spheres (the gumballs) and so I call those the *after* spheres, with radius:

$$r_{\text{after}} = \sqrt{N}. \tag{20}$$

For EcoRI the *before* state is the EcoRI molecule located nonspecifically by electrostatic attraction anywhere on the DNA [56] and the *after* state is EcoRI bound to its specific sites by hydrogen bonds [57] (but not yet cutting the DNA). Likewise, after absorbing a photon, rhodopsin is in the *before* state and it transitions to an *after* state upon dissipation of the photon energy as heat [18, 50, 51].

Following Shannon's lead [21], we can now ask how many small *after* sphere states can fit into the larger *before* sphere state? We find the maximum number by dividing the larger

sphere volume by the smaller sphere volume. Since the volume of a $D$ dimensional sphere is:

$$V = \frac{\pi^{\frac{D}{2}}}{\Gamma\left(\frac{D}{2}+1\right)} r^D \tag{21}$$

[58, 59] the maximum number of *after* sphere states $M$ is:

$$M = \frac{V_{\text{before}}}{V_{\text{after}}} \tag{22}$$

$$= \left(\frac{r_{\text{before}}}{r_{\text{after}}}\right)^D \tag{23}$$

$$= \sqrt{\frac{P}{N}+1}^D \tag{24}$$

using Eqs (19)–(21).

The number of bits needed to select one of the *after* states is the equivalent of the channel capacity, which we could call the biological state capacity:

$$C = \log_2 M \tag{25}$$

$$= \frac{1}{2} D \log_2\left(\frac{P}{N}+1\right) \tag{26}$$

$$(\text{bits per state change}) \tag{27}$$

Since the power $P$ is the energy dissipated for this same *before* to *after* state change, the units are 'joules per state change' and we can now follow Felker and Adler [18, 22, 43–45] by defining the energy needed to gain a bit as:

$$E \equiv \frac{P}{C} \quad (\text{joules per bit}). \tag{28}$$

There is a minimum for $E$ [18] that we can find by taking the limit as $P$ goes to zero using l'Hôpital's rule [60] after substituting Eq (27) into Eq (28):

$$E_{\text{min}} = \lim_{P \to 0} E \quad (\text{joules per bit}) \tag{29}$$

$$= \frac{2\ln 2}{D} \lim_{P \to 0} \frac{P}{\ln(P/N+1)} \tag{30}$$

$$= \frac{2\ln 2}{D} \lim_{P \to 0} \frac{1}{\frac{1}{P/N+1}\frac{1}{N}} \tag{31}$$

$$= \frac{2\ln 2}{D} N \tag{32}$$

Note that if the noise has a thermodynamic origin, then we can substitute Eq (16) into Eq (32) to obtain the thermodynamic form of $E_{\text{min}}$ in Eq (4).

Now we are in a position to define the general Gaussian isothermal efficiency as the minimum energy dissipation per bit ($E_{\min}$) divided by the actual energy dissipation per bit ($E$). We can then derive the efficiency formula by successively substituting in Eqs (27), (28) and (32):

$$\epsilon_t \equiv \frac{E_{\min}}{E} = \frac{\ln\left(\frac{P}{N} + 1\right)}{\frac{P}{N}} \qquad \frac{(\text{joules per bit})}{(\text{joules per bit})}. \tag{33}$$

Remarkably, as with the derivation of the number of states $M$ (Eq (24)), all of the irrelevant constants drop out to leave a pristine 'unitless' isothermal efficiency formula that is only a function of $P/N$. This form is identical to the isothermal efficiency in Eq (9) derived from thermodynamics for communications and molecular machines (Fig 4). However, Eq (33) shows that the result is more general because it only comes from Gaussian distributions. This generalization of the channel capacity and isothermal efficiency works because anytime a Gaussian distribution is observed, it represents noisy energy influencing the system. For example, the roughly Gaussian distribution of MacArthur's famous warbler nesting heights [55] represents the birds having a preferred height ($\mu_i$), but the chosen location of the nest is disturbed by available branches and the energy of wind puffs and wing flutters.

## 4 Discussion

In this paper I have shown that the channel capacity and the isothermal efficiency equations can be derived not only from thermodynamics but more generally from the assumption of Gaussian noise impinging on a high dimensional system. Since Gaussian noise and high dimensionality are found at all levels of biology, these equations can be applied universally to understand biological states.

## Supporting information

**S1 Text. Delila instructions for E. coli LexA binding sites.**
(TXT)

## Acknowledgments

I thank the Advanced Biomedical Computing Center (ABCC) for support, Benjamin Cocanougher, Barry Zeeberg, Susan Carr, Amar Klar, Ilya Lyakhov, Carrie Paterson, Misha Kashlev, André Dormand and John S. Garavelli for useful comments on the manuscript. I also thank Peter J. Thomas for the concept of the formal representation of the molecular state. This paper was presented in part at the meeting Biological and Bio-Inspired Information Theory (14w5170) at the Banff International Research Station (BIRS), Banff, Canada, 2014 Oct 29, 'Three Principles of Biological States: Ecology and Cancer' http://www.birs.ca/events/2014/5-day-workshops/14w5170/videos/watch/201410290904-Schneider.html and at the meeting IEEE Shannon Centenary Day at The Department of Electrical Engineering at the Indian Institute of Technology, Kanpur India, 2016 October 19, 'Information Theory in Biology' http://home.iitk.ac.in/~adrish/Shannon/.

## Author Contributions

**Conceptualization:** Thomas D. Schneider.

**Data curation:** Thomas D. Schneider.

**Formal analysis:** Thomas D. Schneider.

**Funding acquisition:** Thomas D. Schneider.

**Investigation:** Thomas D. Schneider.

**Methodology:** Thomas D. Schneider.

**Project administration:** Thomas D. Schneider.

**Resources:** Thomas D. Schneider.

**Software:** Thomas D. Schneider.

**Supervision:** Thomas D. Schneider.

**Validation:** Thomas D. Schneider.

**Visualization:** Thomas D. Schneider.

**Writing – original draft:** Thomas D. Schneider.

**Writing – review & editing:** Thomas D. Schneider.

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
