## [Decision Letter · Decision Letter 0]

14 Dec 2022

Generalizing the isothermal efficiency by using Gaussian distributions

PONE-D-22-25644

Dear Dr. Schneider,

We’re pleased to inform you that your manuscript has been judged scientifically suitable for publication and will be formally accepted for publication once it meets all outstanding technical requirements.

Kind regards,

Mohammad Mehdi Rashidi

Academic Editor

PLOS ONE

1. Please update your submission to use the PLOS LaTeX template. The template and more information on our requirements for LaTeX submissions can be found at http://journals.plos.org/plosone/s/latex

Reviewers' comments:

Reviewer's Responses to Questions

**Comments to the Author**

1. Is the manuscript technically sound, and do the data support the conclusions?

Reviewer #1: Yes

Reviewer #2: Yes

2. Has the statistical analysis been performed appropriately and rigorously? 

Reviewer #1: Yes

Reviewer #2: Yes

3. Have the authors made all data underlying the findings in their manuscript fully available?

Reviewer #1: Yes

Reviewer #2: Yes

4. Is the manuscript presented in an intelligible fashion and written in standard English?

Reviewer #1: Yes

Reviewer #2: Yes

5. Review Comments to the Author

Reviewer #1: The purpose of this paper is to generalize the efficiency of mathematics for application across biology.

In this paper, they have shown that the channel capacity and the isothermal efficiency equations can be derived not only from thermodynamics but more generally from the assumption of Gaussian noise impinging on a high dimensional system. Since Gaussian noise and high dimensionality are found at all levels of biology, these equations can be applied universally to understand biological states.

Reviewer #2: The manuscript is technically sound and the data support the conclusions. The statistical analysis has been performed appropriately and rigorously. The author has made all data underlying the findings in his manuscript fully available. The manuscript presented in an intelligible fashion and written in standard English.

6. PLOS authors have the option to publish the peer review history of their article (what does this mean?). If published, this will include your full peer review and any attached files.

Reviewer #1: No

Reviewer #2: No

---

## [Editor Report · Acceptance letter]

26 Dec 2022

PONE-D-22-25644 

Generalizing the isothermal efficiency by using Gaussian distributions 

Dear Dr. Schneider:

I'm pleased to inform you that your manuscript has been deemed suitable for publication in PLOS ONE. Congratulations! Your manuscript is now with our production department. 

Kind regards, 

on behalf of

Professor Mohammad Mehdi Rashidi 

Academic Editor

PLOS ONE